# Structure-Based Virtual Screening and De Novo Design to Identify Submicromolar Inhibitors of G2019S Mutant of Leucine-Rich Repeat Kinase 2

**DOI:** 10.3390/ijms232112825

**Published:** 2022-10-24

**Authors:** Hwangseo Park, Taeho Kim, Kewon Kim, Ahyoung Jang, Sungwoo Hong

**Affiliations:** 1Department of Bioscience and Biotechnology, Sejong University, 209 Neungdong-ro, Kwangjin-gu, Seoul 05006, Korea; 2Center for Catalytic Hydrocarbon Functionalizations, Institute for Basic Science (IBS), Daejeon 34141, Korea; 3Department of Chemistry, Korea Advanced Institute of Science and Technology (KAIST), Daejeon 34141, Korea

**Keywords:** virtual screening, de novo design, LRRK2 inhibitor, G2019S mutant, Parkinson’s disease

## Abstract

Missense mutations of leucine-rich repeat kinase 2 (LRRK2), including the G2019S mutant, are responsible for the pathogenesis of Parkinson’s disease. In this work, structure-based virtual screening of a large chemical library was carried out to identify a number of novel inhibitors of the G2019S mutant of LRRK2, the biochemical potencies of which ranged from the low micromolar to the submicromolar level. The discovery of these potent inhibitors was made possible due to the modification of the original protein–ligand binding energy function in order to include an accurate ligand dehydration energy term. The results of extensive molecular docking simulations indicated that the newly identified inhibitors were bound to the ATP-binding site of the G2019S mutant of LRRK2 through the multiple hydrogen bonds with backbone amide groups in the hinge region as well as the hydrophobic interactions with the nonpolar residues in the P-loop, hinge region, and interdomain region. Among 18 inhibitors derived from virtual screening, 4-(2-amino-5-phenylpyrimidin-4-yl)benzene-1,3-diol (Inhibitor **2**) is most likely to serve as a new molecular scaffold to optimize the biochemical potency, because it revealed submicromolar inhibitory activity in spite of its low molecular weight (279.3 amu). Indeed, a highly potent inhibitor (Inhibitor **2n**) of the G2019S mutant was derived via the structure-based de novo design using the structure of Inhibitor **2** as the molecular core. The biochemical potency of Inhibitor **2n** surged to the nanomolar level due to the strengthening of hydrophobic interactions in the ATP-binding site, which were presumably caused by the substitutions of small nonpolar moieties. Due to the high biochemical potency against the G2019S mutant of LRRK2 and the putatively good physicochemical properties, Inhibitor **2n** is anticipated to serve as a new lead compound for the discovery of antiparkinsonian medicines.

## 1. Introduction

Parkinson’s disease (PD) is a neurodegenerative disease that involves a chronic movement disorder in the central nervous system. Due to the lack of a proper therapy for PD progression, the treatments have been limited to the alleviation of the motor symptoms in the early stages of disease. The aberrantly increased activity of leucine-rich repeat kinase 2 (LRRK2) is one of the well-established reasons for the pathogenesis of PD [1,2]. With respect to the pathogenic hyperactive LRRK2 mutation, the missense mutant in which Gly2019 in the activation loop of the kinase domain is replaced with serine is most prevalent in PD patients [3]. Approximately, a 3-fold increase in the autophosphorylation activity of LRRK2 in going from the wild type to the G2019S mutant leads to a significant increase in neurotoxicity [4,5], representing the most frequent risk factor for PD [6]. Furthermore, the effective inhibitors of the G2019S mutant of LRRK2 have the effect of relieving neurodegeneration and thereby decelerating the progression of AD [7,8,9]. The moderation of the abnormal kinase activity of LRRK2 may thus be a promising therapeutic strategy for the treatment of PD.

LRRK2 is a large multifunctional protein comprising 2527 amino acids, which are organized into GTPase and kinase domains along with several protein–protein interacting domains. Although three-dimensional (3D) structures of LRRK2 have been reported in the full-length form [10,11], as well as in part of the kinase domain [12], the lack of structural information about the interactions between the hot-spot residues and a small-molecule ligand has made it challenging to rationally design the potent LRRK2 inhibitors. Nonetheless, structurally diverse inhibitors of LRRK2 have been identified over the past decade. A great deal of scientific endeavors have led to the discovery of novel potent inhibitors involving aminoquinazoline [13], pyrazole biaryl sulfonamide [14], pyrrolo [2,3-d]pyrimidine [15], constrained peptide [16], proteolysis targeting chimera [17], benzothiazole [18], azaindazole [19,20], 4,7-dihydrotetrazolo [1,5-*a*]pyrimidine [21], N-pyridazinylbenzamide [22], 4-ethoxy-7H-pyrrolo [2,3-*d*]pyrimidin-2-amine [23], 3-(4-pyrimidinyl) indazole [24], 2-aminopyridine [25], indolinone [26], triazolopyridazine [27], and 2-anilino-4-methylamino-5-chloropyrimidine [28] as the key structural elements. Although the majority of LRRK2 inhibitors have been identified through the screening of a chemical library or chemical derivatizations of the known inhibitor scaffolds, several rational drug design methods have also been applied using various molecular modeling techniques [29,30,31]. However, clinical applications of the LRRK2 inhibitors have been limited due to the side effects and toxicities as well as to the difficulty in optimizing the brain permeability of ATP-competitive inhibitors [32,33]. To the best of our knowledge, only two small-molecule LRRK2 inhibitors are in clinical trials for the treatment of PD with none being used in clinical treatment [34]. It is thus necessary to find new, effective LRRK2 inhibitors that may develop into a chemotherapeutic drug to prevent the progression of PD.

The present study was undertaken to identify the new LRRK2 inhibitors that can impede the kinase activity of the gain-of-function G2019S mutant. For this purpose, structure-based virtual screening and subsequent biochemical evaluations were carried out extensively for druggable molecules in a commercial compound library. Molecular docking simulations for virtual screening has not always been successful due to the unreliable scoring function to estimate the binding affinity of a putative inhibitor with respect to the target protein [35]. This leads to a low correlation between the computational and experimental results for biochemical potencies. To ameliorate such problematic deviations, the original scoring function was improved by substituting a new ligand dehydration energy term. This augmentation of the scoring function is likely to result in the accuracy enhancement in virtual screening by preventing the underestimation of the ligand dehydration effects in the protein–ligand complexation [36].

To further enhance the performance of virtual screening, a configurational constraint relevant to tight binding to the G2019S mutant was imposed on all the candidate inhibitors screened with docking simulations. More specifically, only the molecules that could presumably form a hydrogen bond with the central amino-acid residue (Ala1950) in the hinge region were considered as the putative inhibitors of the G2019S mutant. This kind of two-step filtration by the binding free energy and the configurational restraint would make the virtual screening more reliable by reducing the false positives produced by the inaccurate protein–ligand binding free energy function. This study will show such stepwise virtual screening to be useful for enriching a chemical library targeted against the G2019S mutant of LRRK2.

## 2. Results and Discussion

The lack of a 3D structure of LRRK2 liganded with a small-molecule inhibitor made it a formidable task to design the novel LRRK2 inhibitors. Therefore, the virtual screening for the discovery of new inhibitors began with the preparation of a proper structural model for the kinase domain of the G2019S mutant using homology modeling. At first, the sequence of 2527 amino acids of human LRRK2 was extracted from the UniProtKB protein knowledge base (http://www.uniprot.org; accession number: Q5S007, accessed on 14 August 2020). The X-ray crystal structure of mixed lineage kinase 1 (MLK1) in complex with a potent inhibitor [37] served as a template to build 3D atomic coordinates of the target protein. The reason for this selection is that MLK1 had the highest score in the amino-acid sequence alignment with the kinase domain of the G2019S mutant (residues 1872-2134) via the basic local alignment search tool (BLAST) [38]. As shown in Figure 1, the identity and similarity between the amino-acid sequences of MLK1 and the G2019S mutant amount to 33.3% and 59.6%, respectively. The structure of the kinase domain of the G2019S mutant seems to be constructed with reasonably good accuracy with the homology modeling because the predicted structure tends to be close to the experimental one when the sequence identity between the template and the target protein is higher than 30% [39].

The homology-modeled structure of the kinase domain of the G2019 mutant was then validated with the ProSa 2003 program, which has been useful for investigating whether the intramolecular interactions of individual amino-acid residues within the entire protein structure would be maintained favorably [40]. This was made possible by calculating the free energy profile for each amino acid using the knowledge-based mean field potential. Figure 2 displays the free energy profile for the homology-modeled structure of the kinase domain of the G2019S mutant in comparison with that of MLK1, which was used as the template for the structural modeling. It is worth noting that the target protein exhibited a better energy profile than the template in the N-terminal and the central region of the kinase domain; although, some relatively unstable regions were also observed. Furthermore, most energy values were maintained as negative, implying that the homology-modeled structure would be physically acceptable. Based on the reasonably good energetic features, the structure of the kinase domain of the G2019S mutant constructed with homology modeling was adopted as the receptor model for the virtual screening to find new LRRK2 inhibitors.

The applicability of molecular docking simulations to virtual screening has been limited by the roughness of the scoring function to calculate the binding affinity of a putative ligand to the target protein [41,42]. In this regard, most popular docking programs tend to underestimate the influence of ligand dehydration energy on protein–ligand binding. This leads to the overestimation of the biochemical potency of hydrophilic molecules [36]. Before virtual screening by docking simulations, therefore, a proper dehydration energy term derived with the extended solvent-contact model was substituted for the original one to enhance the reliability of the scoring function.

Virtual screening was initiated by preparing a chemical library of commercially available molecules that satisfied the criteria of physicochemical properties for druggability. This docking library contained a total of approximately 360,000 ‘Rule-of-Five’-compliant molecules [43], which were selected from 486,035 synthetic and 69,075 natural compounds. Each molecule in the docking library was then screened with docking simulations in the ATP-binding site of the G2019S mutant of LRRK2 to calculate the binding free energy and the binding mode. As shown in Figure 3, a total of 5000 molecules were selected as initial virtual hits in the binding free energy calculation. Keeping in mind that the formation of a hydrogen bond in the hinge region is a common feature of the potent LRRK2 inhibitors [14,15], the molecules likely to establish at least one hydrogen bond with backbone groups in the middle of the hinge region (Ala1950) were considered only for experimental analysis using the associated interatomic distance limit of 3.5 Å. As a result, a total of 429 molecules were selected as final virtual hits for the G2019S mutant of LRRK2. All these putative inhibitors were commercially available from a compound vendor (InterBioScreen Ltd., Bar, Montenegro) and tested for the presence of inhibitory activity against the G2019S mutant using high-throughput binding assays [44]. As a consequence of virtual screening and the subsequent enzyme inhibition assays, 18 compounds were identified as new inhibitors of the G2019S mutant of LRRK2. All these inhibitors exhibited good biochemical potency at submicromolar and low micromolar levels.

The structures and inhibitory activities of 18 new inhibitors of the G2019S mutant of LRRK2 are summarized in Figure 4 and Table 1, respectively, along with the molecular weights (MWs). A common feature of all the new inhibitors (Inhibitors **1**–**18**) is that the hydrogen-bonding moieties are present in the middle of the molecular frameworks while the nonpolar aromatic groups reside in the terminal regions. Therefore, the hydrogen bond and the hydrophobic interactions seem to contribute cooperatively to binding in the ATP-binding site of the G2019S mutant of LRRK2. With respect to excluding the probability of being false positives in the enzyme inhibition assays, Inhibitors **1**–**18** were confirmed as lacking any substructure present in pan assay interference compounds [45]. As can be seen in Table 1, all 18 inhibitors exhibit reasonably high biochemical potencies ranging from the low micromolar level to the submicromolar level. They deserve consideration for further development to enhance biochemical potency and antiparkinsonian activity via structure–activity relationship (SAR) analysis because they were also screened computationally to possess desirable physicochemical properties as drug candidates. Inhibitors **2**, **5**, **9**, and **10** are anticipated, in particular, to serve as a promising molecular core from which nanomolar inhibitors can be derivatized by chemical synthesis because their MWs are lower than 300 amu.

With respect to the structural diversity of the newly discovered inhibitors of the G2019S mutant of LRRK2, Inhibitors **1**–**18** may categorize into 18 molecular cores owing to the lack of any similar chemical moiety among them. Keeping in mind that Inhibitors **1**–**18** are commercial molecules, the structural similarities to the known LRRK2 inhibitors were analyzed using the SciFinder Scholar database. Despite extensive structural searches using the core substructures of Inhibitors **1**–**18** as the input, none of the existing LRRK2 inhibitors were retrieved as the output. This result may provide evidence for the structural novelty of Inhibitors **1**–**18** as new LRRK2 inhibitors.

It is also worth noting that the inhibitory activities of Inhibitors **1**–**6** against the wild type of LRRK2 are comparable to those against the G2019S mutant. This can be attributed to the low MWs of the inhibitors, which makes it possible for the inhibitors to be bound to the two target proteins in a similar fashion. In this regard, all these inhibitors are supposed to reside in close proximity to the hinge region because the formation of at least one hydrogen bond with Ala1950 served as a criterion to select the virtual hits. Because the amino-acid residue 2019 is located on the activation loop and is distant from the hinge region, the mutational status at residue 2019 would have little impact on the binding affinities of Inhibitors **1**–**6** for LRRK2. Hence, more potent and selective G2019S mutant inhibitors than Inhibitors **1**–**6** would be derived by the enlargement of the molecular size in order to facilitate the interactions with the activation loop.

To gain some structural insight into the high inhibitory activities of the newly discovered LRRK2 inhibitors, the patterns of the interactions in the ATP-binding site of the G2019S mutant were addressed in detail. Figure 5 shows the most probable binding modes of Inhibitors **1**–**6** derived from docking simulations using the modified scoring function. Although Inhibitors **1**–**6** appear to be placed extensively in the ATP-binding site, the binding modes exhibit the self-consistency in terms of the interactions with the hot-spot residues. For example, at least one polar group of Inhibitors **1**–**6** points toward the backbone atoms of Ala1950 in the middle of the hinge region to form a hydrogen bond, while the terminal aromatic rings stay in proximity to the glycine-rich phosphate-binding loop (P-loop). This kind of simultaneous binding to the hinge region and the P-loop may explain the submicromolar inhibitory activities of Inhibitors **1**–**6**. Direct interactions with the backbone groups in the hinge region and the hydrophobic sidechains of the P-loop were also implicated in the precedent molecular modeling studies on binding of the potent inhibitors of LRRK2 [13,14,15].

To check the possibility that the newly found LRRK2 inhibitors would bind in a peripheral binding pocket in the non-ATP-competitive way, additional docking simulations of Inhibitors **1**–**6** were carried out with respect to the G2019S mutant of LRRK2. Despite the use of the extensive 3D grid maps to encompass the whole kinase domain, all docking poses produced by 100 docking runs for each inhibitor resided in the ATP-binding site, which implied the lack of a peripheral binding pocket in which Inhibitors **1**–**6** could be more stabilized than in the ATP-binding site. The binding modes of Inhibitors **1**–**6** shown in Figure 5 are also consistent with the dual inhibition of the wild type and the G2019S mutant of LRRK2 (Table 1). Because all the inhibitors reside distant from residue 2019 on the activation loop, the substitution of Ser residue for Gly residue would have little effect on the binding affinity. Thus, the preference of binding in the ATP-binding pocket supports the possibility that Inhibitors **1**–**6** would impair the kinase activity of LRRK2 in an ATP-competitive fashion.

To find the rationales for the high biochemical potencies of the newly identified inhibitors, the calculated binding modes of Inhibitors **1** and **2** in the ATP-binding site of the G2019S mutant were analyzed in detail. The lowest binding energy conformation of Inhibitor **1** with respect to the G2019S mutant is illustrated in Figure 6. We note that that the carbonyl oxygen on the five-membered ring of Inhibitor **1** acts as a hydrogen-bond receptor with respect to the backbone amidic nitrogen of Ala1950. This is consistent with the previous computational finding that the formation of a hydrogen bond with the backbone group of Ala1950 would be necessary for an LRRK2 inhibitor to be bound tightly in the ATP-binding site [13,14,15]. The phenolic oxygen of Inhibitor **1** also receives a hydrogen bond from the sidechain of Arg1957 at the bottom of hinge region. This would in turn facilitate the formation of an additional hydrogen bond with the backbone aminocarbonyl oxygen of Leu1885 that resides on the P-loop. Judging from its capability to form three hydrogen bonds in the ATP-binding site, the 6-hydroxybenzofuran-3-one moiety of Inhibitor **1** would be one of the key structural elements of the LRRK2 inhibitors. Besides the three hydrogen bonds, the nonpolar groups of Inhibitor **1** appear to form van der Waals contacts with the hydrophobic sidechains of Leu1885, Val1893, Phe1890, Ala1904, Met1947, His1998, Leu2001, and Ala2016 in the calculated G2019S–**1** complex. These interactions also seem to be necessary for the high inhibitory activity on the grounds that the associated nonpolar residues reside extensively on the three structural cores of the kinase domain of LRRK2, including the hinge region, P-loop, and activation loop. Taken together, the concurrent establishment of multiple hydrogen bonds and hydrophobic interactions in the ATP-binding pocket may be invoked to elucidate the submicromolar biochemical potency of Inhibitor **1** with respect to the G2019S mutant of LRRK2.

It is worth noting that the inhibitory activities of Inhibitor **2** with respect to the wild type and the G2019S mutant are similar to those of Inhibitor **1** (Table 1) despite the substantial decrease in molecular weight from 401.5 to 279.3 amu. Inhibitor **2** is therefore anticipated to serve as a promising molecular scaffold from which a number of new potent LRRK2 inhibitors can be derivatized by chemical synthesis. Figure 7 shows the most stable binding mode of Inhibitor **2** in the ATP-binding site of the G2019S mutant. It is seen in the calculated G2019S–**2** complex that one of the phenolic moieties adjacent to the central pyrimidin-2-amine group of Inhibitor **2** receives and donates a hydrogen bond from the backbone amidic nitrogen of Ala1950 and to the backbone aminocarbonyl oxygen of Glu1948, respectively. The third hydrogen bond is established between the –NH_2_ moiety on the pyrimidine ring of Inhibitor **2** and the backbone aminocarbonyl oxygen of Ala1950. All these three hydrogen bonds seem to be a significant binding force to stabilize Inhibitor **2** in the ATP-binding site because they involve the backbone groups of the hinge region. In contrast to the same number of hydrogen bonds, Inhibitor **2** occupies less volume than Inhibitor **1** in the ATP-binding pocket and thereby stays more distant from the hydrophobic residues on the P-loop and the activation loop. This would apparently have the effect of weakening the hydrophobic interactions in the G2019S–**2** complex, which is in turn responsible for slightly lower inhibitory activity of Inhibitor **2** than Inhibitor **1**. Nonetheless, Inhibitor **2** deserves the strongest consideration for further development by SAR analysis because of the low molecular weight and the submicromolar inhibitory activity.

With respect to improving the biochemical potency of Inhibitor **2** by synthetic modifications, one of the phenolic moieties needs to be replaced with a nonpolar group because it points toward a small hydrophobic pocket comprising Val1893, Ala1904, and Met1947 (Figure 7). Similarly, the introduction of a nonpolar moiety at the terminal phenyl ring of Inhibitor **2** would also have the effect of enhancing the biochemical potency by strengthening the hydrophobic interactions in the ATP-binding site. The substitution of a nonpolar group has the advantage over the hydrogen-bonding moieties on the grounds that the former has little influence on the dehydration cost whereas introduction of the latter may lead to a decrease in biochemical potency due to the increased stabilization in water. To identify the new inhibitors of the G2019S mutant of LRRK2 with low-nanomolar activity, the structure-based de novo design was carried out using Inhibitor **2** as the molecular core. We tried to find the chemical moieties appropriate for the four substitution positions of the molecular core, which were selected as the derivatization points to enhance the inhibitory activity against the G2019S mutant. This de novo design was proceeded by calculating the binding free energies of varying derivatives of Inhibitor **2** with the core structure being kept fixed.

Among 100 top-scored derivatives of Inhibitor **2** generated in the de novo design, 16 were commercially available and tested for biochemical potency by enzyme inhibition assays. Table 2 lists the chemical structures and the inhibitory activities of the new G2019S mutant inhibitors derived from Inhibitor **2**. We note that most derivatives with good biochemical potency prefer the hydrogen atom or only the small substituents at R_2_, R_3_, and R_4_ positions, while relatively bulky groups such as long alkyl chains and aromatic rings are allowed at the R_1_ position of Inhibitor **2**. For example, new inhibitors more potent than Inhibitor **2** could be identified by a single substitution of the ethyl moiety at the R_1_ position (Inhibitor **2b**) as well as by double substitutions of methyl and methoxy groups at the R_1_ and R_4_ positions (Inhibitor **2m**), respectively. The inhibitory activity surges to the nanomolar level by enlarging the R_1_ substituent from methyl in Inhibitor **2m** to the ethyl moiety in Inhibitor **2n**. However, the introduction of the substituents bulkier than the ethyl group causes a decrease in the biochemical potency in all cases. Because Inhibitor **2n** was also computationally screened for possessing the desirable physicochemical properties as a drug candidate, it deserves consideration for further investigation to develop antiparkinsonian medicines.

To find the reason for the nanomolar-level inhibitory activity of Inhibitor **2n**, the binding mode in the ATP-binding site of the G2019S mutant of LRRK2 was investigated with docking simulations. The highest-score binding configurations is illustrated in Figure 8 in comparison with that of DNL-151, a nanomolar LRRK2 inhibitor in clinical trials. The calculated binding mode of DNL-151 is similar to that of Inhibitor **2n** in that the two hydrogen bonds with backbone groups of Ala1950 in the middle of the hinge region play a dominant role in the complexation. Inhibitor **2n** appears to bind in the ATP-binding pocket almost in the same fashion as Inhibitor **2** in terms of the hydrogen-bond interactions. Due to the respective substitutions of the ethyl and methoxy moieties at the R_1_ and R_4_ positions, however, Inhibitor **2n** gets closer to Val1893 on the P-loop and Met1947 in the hinge region as well as to Leu2001 at the bottom of the ATP-binding site. It is thus apparent that the enhancement of the biochemical potency from the submicromolar level in Inhibitor **2** to the nanomolar level in Inhibitor **2n** can be attributed to the strengthening of hydrophobic interactions with the G2019S mutant of LRRK2.

A common structural feature found in the G2019S–**1**, G2019S–**2**, and G2019S–**2n** complexes is that the hydrophobic contacts are present in proximity to the intermolecular hydrogen bonds. In fact, the hydrophobic interactions would have the effect of strengthening the neighboring hydrogen bonds by limiting the approach of hydrolytic solvent molecules, leading to a synergistic effect on the biochemical potency. Indeed, proper positioning of the hydrophobic contacts in the vicinity of the hydrogen bonds has often served as a facile strategy to optimize the protein–ligand binding affinity [46,47].

We attempted to address the dynamic properties of the G2019S–**2n** complex to elucidate the nanomolar-level biochemical potency of Inhibitor **2n** by conducting molecular dynamics (MD) simulations in aqueous solution. The dynamic stability of the G2019S mutant of LRRK2 was estimated by the time evolutions of the root-mean-square deviation (RMSD_init_) of the backbone C_α_ atoms from the initial structure of the G2019S–**2n** complex. As can be seen in Figure 9, the RMSD_init_ values are within 2.5 Å during the entire course of the simulation and become convergent with respect to the simulation time. These results imply that the structure of the G2019S mutant of LRRK2 would maintain stability without a significant conformational change. It is also worth noting that the RMSD_init_ values of the heavy atoms of Inhibitor **2n** fall into 0.6 Å and remain even lower than those of the C_α_ atoms of the G2019S mutant throughout the simulation time. This suggests that the positional shifts of Inhibitor **2n** in the ATP-binding site are insignificant in comparison to the conformational changes of the G2019S mutant of LRRK2, which is consistent with the nanomolar-level inhibitory activity of Inhibitor **2n**.

Although structurally diverse and potent inhibitors of the G2019S mutant of LRRK2 were identified using extensive virtual screening and a de novo design, it was difficult to raise the biochemical potencies to the low nanomolar level. This indicates that the scoring function remains imperfect despite modification with a new dehydration term. Such flaws of the scoring function can be attributed in a large part to the inappropriate optimization of the weighting factors for the five energy terms, which was actually inevitable because no LRRK2 inhibitor was included in the training set for parameterizations. Hence, the scoring function is expected to become more reliable by reoptimizing the weighting factors using a new training set augmented by a variety of LRRK2 inhibitors along with the associated K_i_ values. Such an accuracy enhancement would also facilitate the design of the selective inhibitors of the G2019S mutant over the wild type. This can be fueled by the experimental data available for a variety of selective G2019S mutant inhibitors. Future studies will be focused on identifying the selective low-nanomolar inhibitors of the G2019S mutant of LRRK2 via docking simulations and de novo design with a further modified scoring function.

## 3. Materials and Methods

### 3.1. Structural Preparations of the G2019S Mutant of LRRK2

The receptor model for the G2019S mutant of LRRK2 was prepared from homology modeling using the X-ray structure of MLK1 in complex with a potent inhibitor of nanomolar-level activity (PDB code: 3DTC) [37] as the structural template. The latest version (10.2) of the MODELLER program was used in this structural construction [48]. To obtain the 3D atomic coordinates of the G2019S mutant of LRRK2, we employed the optimization algorithm of the conjugate gradient method along with molecular dynamics simulations to minimize the violations of spatial restraints. The final structural model for the G2019S mutant served as the starting point for virtual screening of the inhibitors from a large commercial chemical library.

To obtain the all-atom model for the receptor protein, hydrogen atoms were added to each heavy atom of the G2019S mutant. For this purpose, the protonation states of all the titratable residues (Asp, Glu, His, and Lys) were determined carefully according to the hydrogen-bonding patterns in the homology-modeled structure of the G2019S–inhibitor complex. For instance, the sidechains of the Asp and Glu residues were assumed to be neutral if either of the carboxylate oxygens pointed toward a hydrogen-bond acceptor atom within the distance limit of 3.5 Å. Similarly, the sidechains of lysine were considered positively charged unless the amine moiety resided in the vicinity of a hydrogen-bond donor atom. The same criterion was also used for determining the protonation states of all histidine sidechains.

### 3.2. Structure-Based Virtual Screening to Identify the Inhibitors of the G2019S Mutant of LRRK2

Prior to virtual screening with molecular docking simulations, we constructed a docking library of the G2019S mutant comprising approximately 360,000 synthetic and natural compounds from the latest version (February 2022) of the chemical database distributed by InterBioScreen Ltd. (Bar, Montenegro). This was made possible by screening 560,000 compounds in the original chemical database based on Lipinski’s “Rule of Five” to select only the molecules that possessed the physicochemical properties desirable for potential drug candidates [43]. All these screened molecules were used as the inputs of the CORINA program to generate their 3D atomic coordinates, followed by the calculations of atomic charges with the Gasteiger–Marsilli method [49]. Finally, all molecules in the docking library were virtually screened with docking simulations using the modified version of the AutoDock program [50] to find the actual inhibitors of the G2019S mutant of LRRK2.

Although ligand hydration effects have a critical effect on the protein–ligand association [51], the scoring function of the original AutoDock program includes a rough form of the hydration energy term in which only six atom types are taken into account. Therefore, a new scoring function was derived to increase the accuracy of virtual screening by replacing the original dehydration energy term with the modified one. This new scoring function (Δ*G_b_^aq^*) can be expressed as follows:(1)$$\begin{array}{ll} \Delta G_{b}^{aq} & =W_{vdW}\sum_{i=1}\sum_{j=1}\left(\frac{A_{ij}}{r_{ij}^{12}}-\frac{B_{ij}}{r_{ij}^{6}}\right)+W_{hbond}\sum_{i=1}\sum_{j=1}E(t)\left(\frac{C_{ij}}{r_{ij}^{12}}-\frac{D_{ij}}{r_{ij}^{10}}\right)\\ & +W_{elec}\sum_{i=1}\sum_{j=1}\frac{q_{i}q_{j}}{\varepsilon (r_{ij})r_{ij}}+W_{tor}N_{tor}+W_{sol}\sum_{i=1}S_{i}\left(Occ_{i}^{\max}-\sum_{j\neq i}V_{j}e^{-\frac{r_{ij}^{2}}{2\sigma^{2}}}\right) \end{array}$$
where *W_vdW_*, *W_hbond_*, *W_elec_*, *W_tor_*, and *W_sol_* denote the weighting factors of the van der Waals interaction, hydrogen bond, electrostatic interaction, torsional motion, and dehydration energy of a putative inhibitor, respectively. *r_ij_* indicates the interatomic distance, and *A_ij_*, *B**_ij_*, *C**_ij_*, and *D_ij_* are associated with the well depth and the equilibrium distance in the potential energy function for protein–ligand atom pairs. AMBER force field parameters were used in calculating all van der Waals interaction energies as implemented in the original AutoDock program. The hydrogen-bond term includes the additional weighting factor (*E*(*t*)) to represent the angle-dependent directionality. To calculate the interatomic electrostatic interaction energies between LRRK2 and a putative inhibitor, the sigmoidal function of *r_ij_* proposed by Mehler et al. was used as the distance-dependent dielectric constant [52]. In the torsional energy term, *N_tor_* refers to the number of rotatable bonds in a molecule. In the dehydration energy term, *S_i_* and *V_i_* stand for the atomic solvation energy per unit volume and the fragmental atomic volume, respectively, while *Occ_i_*^max^ indicates the maximum occupancy of an atom in a candidate inhibitor [53]. The atomic parameters derived from the solvent-contact model were used in calculating the dehydration energies of all the molecules in the docking library as they had shown a good performance in blind prediction challenges [54,55]. The incorporation of this new dehydration energy term is most likely to enhance the reliability of the protein–ligand binding energy function because the biochemical potency of a polar chemical moiety tends to be overestimated when the ligand hydration effects are underestimated [36]. Using the new scoring function, molecular docking simulations were carried out in the ATP-binding site of the G2019S mutant of LRRK2 to score and rank the candidate inhibitors according to the calculated binding affinities.

### 3.3. De Novo Design to Enhance the Biochemical Potency

The structure-based de novo design was also applied in this work to find the derivatives of Inhibitor **2** that would have enhanced inhibitory activity against the G2019S mutant as well as good physicochemical properties to be potential drug candidates. The LigBuilder program [56] was employed in this de novo design using the structure of the G2019S–**2** complex calculated with docking simulations. In the first step, the empty space in the ATP-binding site of the G2019S mutant was explored to find the space for accommodating the derivatives of Inhibitor **2**. More specifically, the structure of the G2019S mutant in complex with Inhibitor **2** served as the input to find the key interaction residues in the ATP-binding site. The next step involved the generation of the derivatives of Inhibitor **2** by applying the genetic algorithm and the bioavailability rules to select only the derivatives with good physicochemical properties to be potential drug candidates. The structure of Inhibitor **2** was thus evolved from the hydrogens attached to the four substitution points of the molecular core. To reduce the computational burden for finding the optimal binding modes and the binding affinities, only the substituents were allowed to move in the ATP-binding site while the core structure was kept fixed. A total of 100,000 derivatives were generated and scored according to the binding affinities calculated with the modified scoring function. The 100 top-scored molecules were then checked for commercial availability. Finally, 16 derivatives of Inhibitor **2** were purchased from a chemical vendor and tested for inhibitory activity against the G2019S mutant.

### 3.4. Molecular Dynamics Simulations

To investigate the dynamic properties of the G2019S mutant of LRRK2 in complex with a potent inhibitor, we carried out MD simulations in aqueous solution. In the preliminary step, the structure of the G2019S–**2n** complex derived from docking simulations was equilibrated with the AMBER program, which has been widely used in modeling the structures and functions of biomolecules in solution [57]. The solute system was augmented by the three Na^+^ ions as the counter ions to neutralize the total charge of the all-atom model for the G2019S mutant of LRRK2. The solute system comprising the G2019S mutant of LRRK2, Inhibitor **2n**, and three Na^+^ ions was then immersed in a rectangular solvent box containing 11,433 TIP3P [58] water molecules. After 1000 cycles of energy minimization to remove the physically unacceptable contacts, we equilibrated the system beginning with 20 picosecond (ps) equilibration dynamics of the water molecules at 300 K. The next step was the equilibration of the solute in a fixed configuration of water molecules for 10 ps at 10, 50, 100, 150, 200, 250, and 300 K. This was followed by the equilibration dynamics of the entire system at 300 K for 500 ps using the periodic boundary condition. The SHAKE algorithm [59] was used with a tolerance of 10^−6^ Å to fix all bond lengths involving a hydrogen atom. Finally, 10.2 nanosecond production dynamics simulations were performed with periodic boundary conditions in the NPT ensemble. The temperature and pressure were kept at 300 K and 1 atm using Berendsen temperature coupling [60] and isotropic molecule-based scaling, respectively. We used the time step of 2.0 femtosecond and the nonbond-interaction cut-off radius of 12 Å. The trajectory was sampled every 0.4 ps (200-step intervals) for analysis.

### 3.5. Enzyme Inhibition Assays

All enzyme inhibition assays were performed using the radiometric ([γ-^33^P]-ATP) kinase assays from Reaction Biology Corp. (Malvern, PA, USA). A broad-spectrum kinase inhibitor (staurosporine) was used as the positive control. Each candidate inhibitor was tested in a buffer containing Brij-35 detergent, a nonionic polyoxyethylene surfactant, to avoid aggregate formation. Among a total of 429 virtual hit molecules found in virtual screening, those that impeded the kinase activity of the G2019S mutant more than 50% at 10 μM were selected to determine the IC_50_ values. For each inhibitor, the inhibitory activities required to calculate the IC_50_ value were measured at ten different concentrations, which were monitored by the percentage of the remaining kinase activity with respect to the vehicle reaction by solvent molecules (dimethyl sulfoxide). The IC_50_ value of each inhibitor was then derived with the curve fits (Appendix A) produced by the PRISM program (GraphPad Software, San Diego, CA, USA).

## 4. Conclusions

By means of virtual screening with molecular docking simulations, we identified 18 inhibitors of the G2019S mutant of LRRK2, the biochemical potencies of which ranged from the low micromolar to the submicromolar level. This accomplishment was made possible by the modification of the protein–ligand binding free energy function in order to involve a proper ligand dehydration energy term. The results of extensive docking simulations indicated that the potent inhibitors are accommodated in the ATP-binding pocket of the G2019S mutant through the multiple hydrogen bonds with backbone amide groups in the hinge region, along with the hydrophobic contacts with nonpolar residues in the P-loop, hinge region, and interdomain region. Of 18 inhibitors derived from virtual screening, 4-(2-amino-5-phenylpyrimidin-4-yl)benzene-1,3-diol (Inhibitor **2**) was most likely to serve as a new molecular scaffold to optimize the biochemical potency because it revealed submicromolar inhibitory activity in spite of its lowest molecular weight (279.3 amu). Indeed, a highly potent inhibitor of the G2019S mutant with nanomolar-level activity was also discovered via the structure-based de novo design using the structure of Inhibitor **2** as the molecular scaffold. This potency enhancement could be attributed to the strengthening of hydrophobic interactions caused by the substitutions of small nonpolar moieties. Due to the high inhibitory activity against the G2019S mutant of LRRK2 and the putatively good physicochemical properties, Inhibitor **2n** is anticipated to serve as a new lead compound for the discovery of antiparkinsonian medicines.

## Figures and Tables

**Figure 1 ijms-23-12825-f001:**
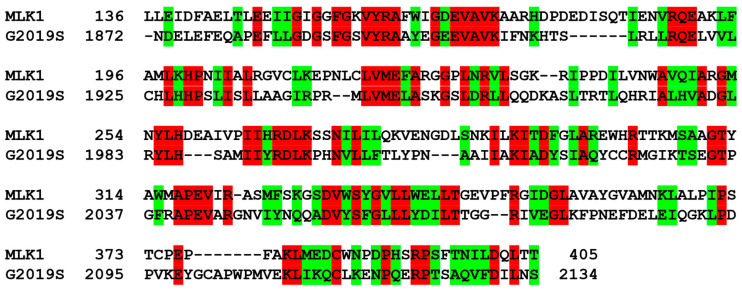
Amino-acid sequence alignment between the kinase domains of MLK1 and the G2019S mutant of LRRK2. Identical and similar residues are colored in red and green, respectively.

**Figure 2 ijms-23-12825-f002:**
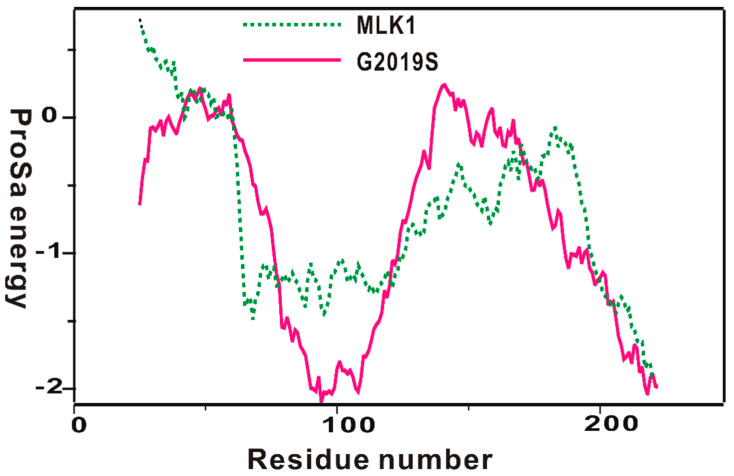
Comparative view of the ProSa energy profile for the homology-modeled structure of the G2019S mutant of LRRK2 and that of the template. The amino acids of the two proteins were renumbered from 1 for convenience.

**Figure 3 ijms-23-12825-f003:**
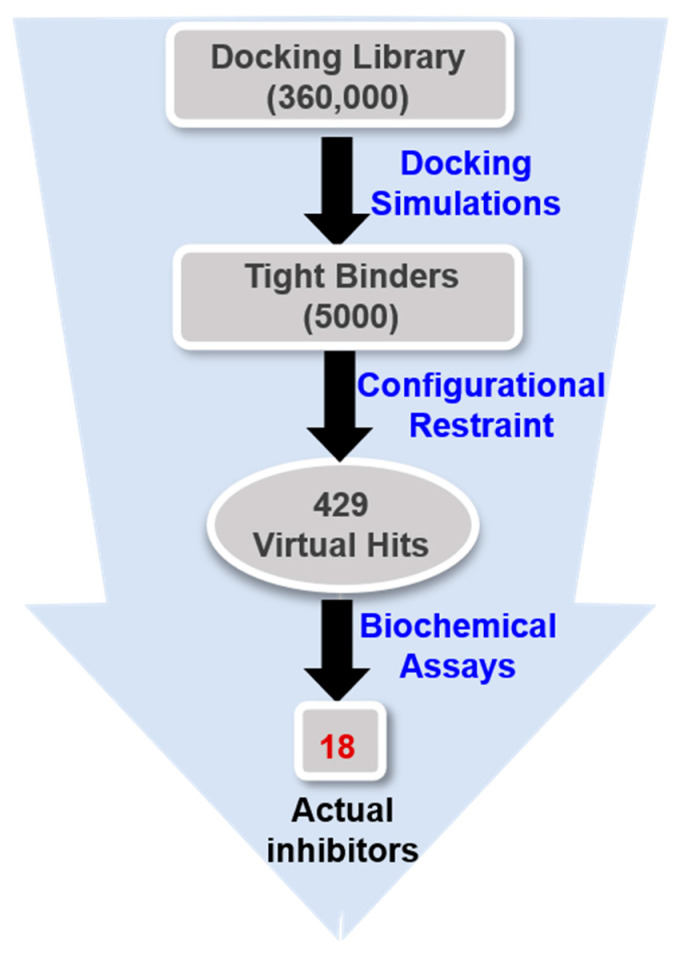
Flowchart for the discovery of new small-molecule inhibitors of the G2019S mutant of LRRK2 through virtual screening and enzyme inhibition assays.

**Figure 4 ijms-23-12825-f004:**
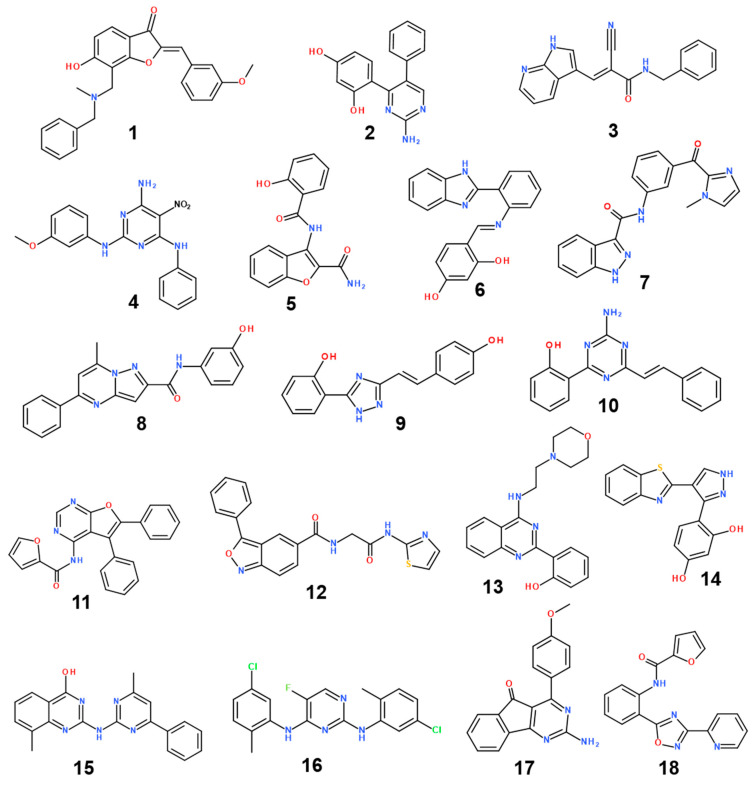
Chemical structures of 18 new inhibitors of the G2019S mutant of LRRK2.

**Figure 5 ijms-23-12825-f005:**
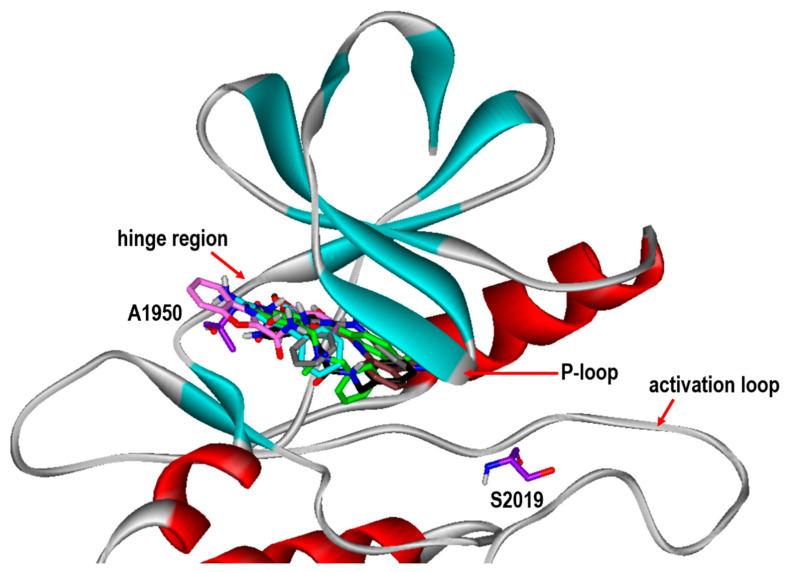
Comparative view of the binding modes of Inhibitors **1-****6** in the ATP-binding sites of the G2019S mutant of LRRK2. The carbon atoms of LRRK2 and Inhibitors **1-6** are indicated in violet, green, cyan, black, gray, pink, and brown, respectively. The positions of Ala1950 in the hinge region and mutated Ser2019 are also indicated.

**Figure 6 ijms-23-12825-f006:**
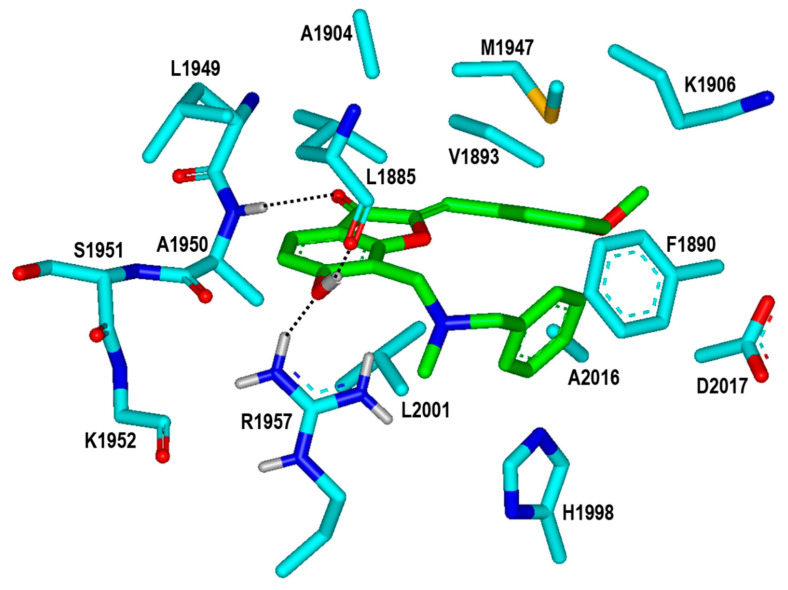
Calculated binding mode of Inhibitor **1** in the ATP-binding site of the G2019S mutant of LRRK2. The carbon atoms of the G2019S mutant and Inhibitor **1** are colored in cyan and green, respectively. Each dotted line indicates a hydrogen bond.

**Figure 7 ijms-23-12825-f007:**
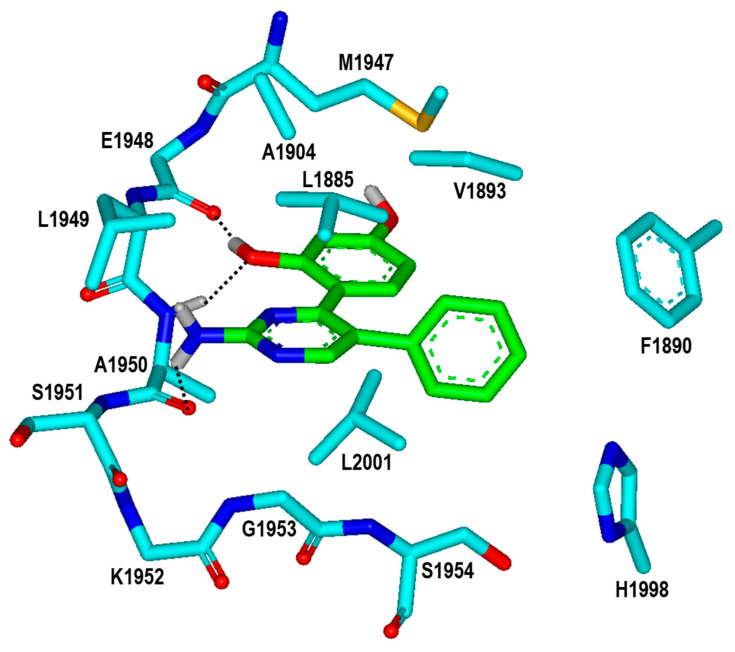
Calculated binding mode of Inhibitor **2** in the ATP-binding site of the G2019S mutant of LRRK2. The carbon atoms of the G2019S mutant and Inhibitor **2** are colored in cyan and green, respectively. Each dotted line indicates a hydrogen bond.

**Figure 8 ijms-23-12825-f008:**
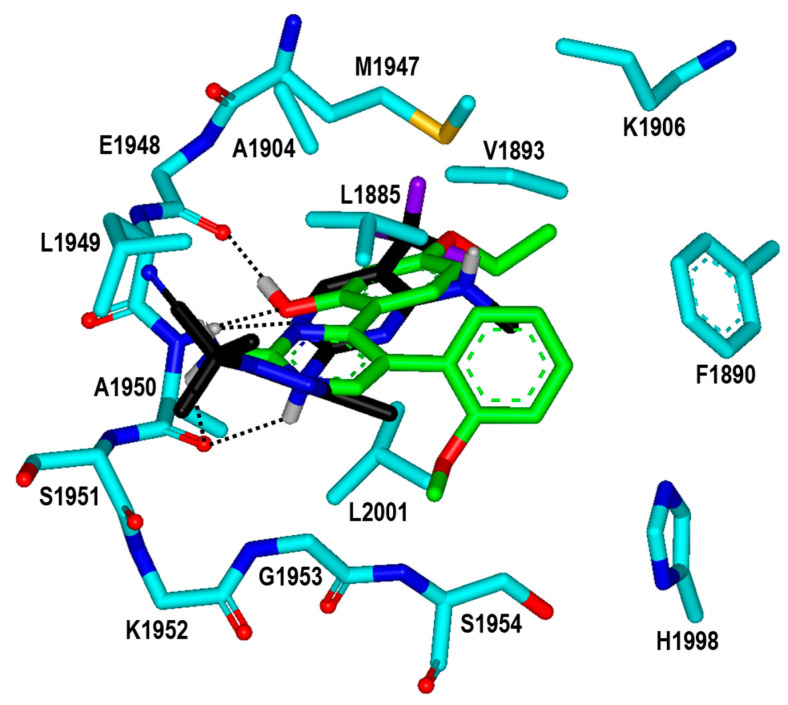
Calculated binding mode of Inhibitor **2n** in the ATP-binding site of the G2019S mutant of LRRK2 in comparison with that of DNL-151. The carbon atoms of the protein, Inhibitor **2n**, and DNL-151 are indicated in cyan, green, and black, respectively. Each dotted line indicates a hydrogen bond.

**Figure 9 ijms-23-12825-f009:**
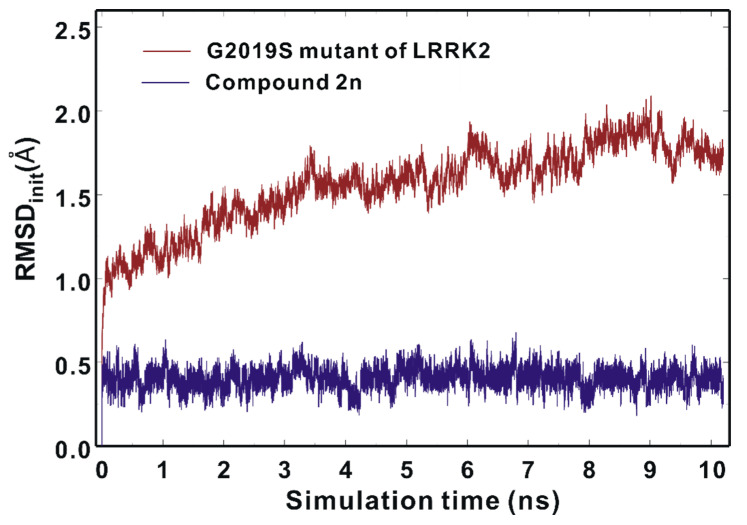
Time evolutions of the root-mean-square deviation (RMSD_init_) of the backbone C_α_ atoms of the G2019S mutant (brown) and those of the heavy atoms of Inhibitor **2n** (blue).

**Table 1 ijms-23-12825-t001:** Molecular weight (MW), calculated binding free energy (*Δ**G_bind_*), and IC_50_ (in μM) values of 18 new inhibitors of the G2019S mutant of LRRK2.

Inhibitor	MW (amu)	Δ*G_bind_* (kcal/mol)	IC_50_ (μM)	
			G2019S Mutant	Wild Type
**1**	401.5	−26.16	0.194 ± 0.017	0.187 ± 0.005
**2**	279.3	−26.02	0.203 ± 0.007	0.260 ± 0.005
**3**	316.4	−26.03	0.206 ± 0.029	0.301 ± 0.028
**4**	352.4	−26.44	0.244 ± 0.016	0.261 ± 0.017
**5**	296.3	−26.08	0.276 ± 0.032	0.235 ± 0.018
**6**	329.4	−25.91	0.289 ± 0.011	0.254 ± 0.007
**7**	345.4	−26.02	0.413 ± 0.008	ND
**8**	344.4	−26.98	0.494 ± 0.028	ND
**9**	279.3	−25.91	0.505 ± 0.064	ND
**10**	290.3	−26.31	0.563 ± 0.019	ND
**11**	381.4	−26.86	0.652 ± 0.033	ND
**12**	378.4	−27.53	0.681 ± 0.024	ND
**13**	350.4	−26.92	0.776 ± 0.027	ND
**14**	309.4	−26.17	1.918 ± 0.051	ND
**15**	343.4	−27.01	1.971 ± 0.313	ND
**16**	377.3	−26.14	4.408 ± 0.053	ND
**17**	303.3	−27.24	4.464 ± 0.024	ND
**18**	345.4	−26.42	8.942 ± 0.067	ND

**Table 2 ijms-23-12825-t002:** Structures and inhibitory activities of the derivatives of Inhibitor **2** against the G2019S mutant.

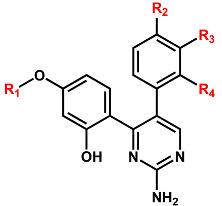
	R_1_ *^a^*	R_2_	R_3_	R_4_	IC_50_ (μM)
**2a**	CH_3_	H	H	H	0.400 ± 0.011
**2b**	CH_3_CH_2_	H	H	H	0.123 ± 0.007
**2c**	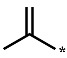	H	H	H	0.108 ± 0.005
**2d**	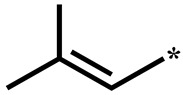	H	H	H	0.762 ± 0.025
**2e**	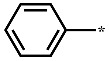	H	H	H	2.627 ± 0.304
**2f**	CH_3_	CH_3_O	H	H	0.479 ± 0.009
**2g**	CH_3_CH_2_	Cl	H	H	1.115 ± 0.130
**2h**	CH_3_CH_2_	Br	H	H	1.550 ± 0.213
**2i**	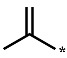	CH_3_O	H	H	1.353 ± 0.062
**2j**	CH_3_	CH_3_O	CH_3_O	H	0.883 ± 0.079
**2k**	CH_3_CH_2_	CH_3_O	CH_3_O	H	0.438 ± 0.023
**2l**	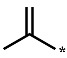	CH_3_O	CH_3_O	H	1.823 ± 0.187
**2m**	CH_3_	H	H	CH_3_O	0.122 ± 0.008
**2n**	CH_3_CH_2_	H	H	CH_3_O	0.089 ± 0.006
**2o**	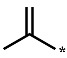	H	H	CH_3_O	0.316 ± 0.037
**2p**	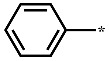	H	H	CH_3_O	0.613 ± 0.056

*^a^* Asterisk indicates the atom attached to the position of substitution.

## Data Availability

All molecular docking simulations were carried out with the open-source package AutoDock, which can be downloaded free of charge at https://autodock.scripps.edu. The chemical database used for identifying the inhibitors of the G2019S mutant LRRK2 was obtained from InterBioScreen Ltd., and is freely distributed at https://www.ibscreen.com/.

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
