# Peer review of "Structure-Based Virtual Screening and De Novo Design to Identify Submicromolar Inhibitors of G2019S Mutant of Leucine-Rich Repeat Kinase 2"

_ijms, 2022, doi:10.3390/ijms232112825_

Round 1
Reviewer 1 Report
In this study, the authors presented the results of structure-based virtual screening of a large chemical library to identify some novel inhibitors of the 15 G2019S mutant LRRK2. The modified scoring function was derived to increase the accuracy. Also, for some promising inhibitor candidates, inhibition assay tests were performed. The manuscript is written correctly, the methods are well-chosen, and the results are clearly presented. This study is interesting. The obtained results provide a solid and promising ground for designing new compounds as a potential treatment for Parkinson’s disease. I have no negative comments. Only one minor:
Tables 1 and 2 – the standard deviation (SD) of IC50 should be added (from the fitted data in Prism)
Author Response
Following the suggestion, we have added the standard deviations for the IC50 values of the newly identified LRRK2 inhibitors in Tables 1 and 2.
Reviewer 2 Report
The paper describe the classical way of the search for the ATP-concurrent small molecule inhibitors: a library screening and structural optimization of the best compound. However, I think the idea to use the homology model instead of the crystal structure is discussible.
More important, that the field of research chosen was actively developed during a several decades. Thus in 2022, in my opinion, if the author wish to publish their work, they have to provide some additional data, without which the results are not conclusive.
The paper missed:
1) The calculated binding affinities of the inhibitors presented (as well as that of ATP), and the discussion of the concurrentability of the inhibitors.
2) The inhibitory curves for the best compounds, which allowed to calculated the submicromolar IC50
In addition,
3) Not only static position of the best inhibitors in the ATP-binding pocket, but their dynamics should be provided.
4) The issue of selectivity should be addressed somehow.
5) The comparison of the new compounds with the previously published LRRK2 inhibitors (at least with those in the clinical trials ) are welcome.
6) The Methods section should be divided according to the different type of the research.
Author Response
1) This paper missed the calculated binding affinities of the inhibitors presented (as well as that of ATP), and the discussion of the concurrentability of the inhibitors.
In accordance with the comment, we have presented the calculated binding free energies for the newly identified LRRK inhibitors in Table 1. With respect to the concurrentability of the hydrogen bonds and hydrophobic interactions in inhibitor binding, it is found to be a common structural feature in G2019S-1, G2019S-2, and G2019S-2n complexes that the hydrophobic contacts are present in proximity to the intermolecular hydrogen bonds. Actually, the hydrophobic interactions would have the effect of strengthening the neighboring hydrogen bonds by limiting the approach of hydrolytic solvent molecules, leading to the synergistic effect on the biochemical potency. Indeed, a proper positioning of the hydrophobic contacts in the vicinity of the hydrogen bonds has often served as a facile strategy to optimize the protein-ligand binding affinity. To explain these, we have added three sentences at the top of p. 20 in the revised manuscript, along the two references (Refs. 46 and 47).
2) This paper missed the inhibitory curves for the best compounds, which allowed to calculate the submicromolar IC50.
The dose-response curve fits of 1-18 and 2a-2p, which were used to obtain the IC50 values, have been presented in Supplementary Materials.
3) Not only static position of the best inhibitors in the ATP-binding pocket, but their dynamics should be provided.
Following the suggestion, we have performed molecular dynamics (MD) simulations to address the dynamic properties of G2019S-2n complex in aqueous solution. The dynamic stability of the G2019S mutant LRRK2 was estimated by the time evolutions of the root mean square deviation (RMSD) of backbone Ca atoms from the initial structure of G2019S-2n complex. The RMSD values were within 2.5 Å during the entire course of simulation, which become convergent with respect to the simulation time. These results implied that the structure of G2019S mutant LRRK2 would be maintained stable without a significant conformational change. It was also worth noting that the RMSD values of the heavy atoms of 2n fell into 0.6 Å and remained even lower than those of Ca atoms of the G2019S mutant throughout the simulation time. This suggested that the positional shifts of 2n in the ATP-binding site were insignificant in comparison to the conformational changes of the G2019S mutant LRRK2, which was consistent with the nanomolar-level inhibitory activity of 2n. To present and discuss the results of MD simulations, we have added a paragraph at the end of p. 20 in the revised manuscript along with a figure (Figure 9).
4) The issue of selectivity should be addressed somehow.
With respect to discovering the selective inhibitors of G2019S mutant over the wild type, the further enhancement of the scoring function is necessary because both amino acid sequences and structures of the wild type and the mutant are almost the same. Designing the selective inhibitors can be fueled by the experimental data available for a variety of selective G2019S mutant inhibitors. To place an emphasis on this point, we have added two sentences on p. 21 line 6 from bottom in the revised manuscript.
5) The comparison of the new compounds with the previously published LRRK2 inhibitors (at least with those in the clinical trials) are welcome.
In accordance with the comment, the binding mode of the most potent inhibitor (2n) was analyzed in comparison with that of DNL-151, a nanomolar LRRK2 inhibitor in clinical trials. The calculated binding mode of DNL-151 is similar to that of 2n in that the two hydrogen bonds with backbone groups of Ala1950 in the middle of the hinge region play a dominant role in the complexation. These newly obtained results have been presented and discussed on p. 19 in the revised manuscript.
6) The Methods section should be divided according to the different type of the research.
As requested, the Materials and Methods section has been divided into the four subsections according to the type of computational and experimental approaches.
Round 2
Reviewer 2 Report
In tables where IC50 are presented, it is enough to leave 2 decimal places.